# Symptom profiles in lung cancer survivors: A latent class approach

**Tehreem Hussain[1], Elisa H. Son[2], Gwenyth R. Wallen[2], Li Yang[2], Lena J. Lee[2]***

**1** Emory University Rollins School of Public Health, Atlanta, Georgia, United States of America, **2** National Institutes of Health Clinical Center, Translational Biobehavioral and Health Promotion (TBHP), Bethesda, Maryland, United States of America

* jumin.park@nih.gov

## Abstract

Lung cancer survivors experience multiple concurrent symptoms after cancer treatments. However, the majority of symptom research has focused on assessing and managing individual symptoms. Furthermore, little is known about the risk factors and adverse outcomes of complex symptoms in lung cancer survivors. The purpose of the study was to: (1) identify symptom profiles in lung cancer survivors; (2) determine influencing factors of the symptom profiles; and (3) examine differences in health outcomes among the symptom profiles. A cross-sectional secondary analysis of data from the Measuring Your Health (MY-Health) study was conducted with 526 lung cancer survivors. Symptom profiles were identified using latent profile analysis based on four patient-reported symptoms (pain, fatigue, sleep disturbance, and depression) with custom PROMIS® short forms. We conducted multinomial logistic regression analysis to determine influencing factors of the symptom profiles and multivariate analysis of variance to examine differences in physical function and social function among the symptom profiles. Four latent class symptom profiles were identified: (1) *Within Normal Limits* (Class 1), (2) *Pain with Fatigue and Sleep Disturbance* (Class 2), (3) *Depression with Fatigue and Sleep disturbance* (Class 3), and (4) *All High Symptom Burden* (Class 4). Age, income, employment status, and number of comorbidities were the influencing factors of the symptom profiles. There were significant differences in physical function and social function among the symptom profiles. This study found that the influencing factors of the symptom profiles in lung cancer survivors tended to be more sociodemographic in nature, rather than clinical. Researchers and healthcare providers use findings such as these when establishing symptom management strategies for lung cancer survivors by integrating demographic and socioeconomic determinants of health in conjunction with targeted clinical variables.

**Data availability statement:** All data files are available from the HealthMeasures Dataverse at https://dataverse.harvard.edu/dataverse/HealthMeasures.

**Funding:** This research was supported by the Intramural Research Program of the National Institutes of Health (NIH). The contributions of the NIH author(s) are considered Works of the United States Government. The findings and conclusions presented in this paper are those of the author(s) and do not necessarily reflect the views of the NIH or the U.S. Department of Health and Human Services.

**Competing interests:** The authors have declared that no competing interests exist.

## Introduction

Lung and bronchus cancer are associated with the highest cancer-related mortality rate; in 2024, 125,070 patients were expected to die just in the United States due to lung and bronchus cancer [1]. The five-year lung cancer survival rate between 2014 and 2020 was 26.7%, [1]. For lung cancer survivors, symptoms such as pain, fatigue, psychological issues, and sleep disturbance can be traced across different studies [2–4]. The symptom burden experienced by lung cancer survivors may lead to poor performance, such as a decrease in physical function or social function [5,6].

Although these symptoms in lung cancer survivors tend to occur concurrently, they have often been studied and managed individually [5–7]. The co-occurrence of multiple symptoms is called symptom clusters, defined as: "two or more symptoms that are related to each other and that occur together." [8] Symptom clusters consist of stable subgroups of symptoms; each subgroup is relatively independent of the other subgroups and may reveal specific underlying dimensions of symptoms [8,9]. Symptom clusters have been identified using two main approaches: variable-centered approaches and patient-centered approaches [10]. Variable-centered approaches identify symptoms that empirically cluster together though analytic methods (e.g., factor analysis), which create distinct groups of related symptoms (i.e., symptom clusters). Patient-centered approaches (e.g., latent class analysis) identify subgroups of patients with distinct symptom profiles based on one or more symptoms or a pre-specified symptom cluster (e.g., pain, fatigue, sleep disturbance, depression). Latent class analysis is a statistical method to identify unobserved class membership based on observed indicators, which can be used to explore symptom profiles in a given sample [11]. Latent class analysis is a person-centered analytic approach that focuses on grouping individuals who share similar characteristics [12]. Building on this framework, growing evidence suggests that psychoneurological symptoms (e.g., pain, fatigue, sleep disturbance, depression) often co-occur, forming a cluster that may reflect underlying psychological and/or neurological dysfunction and is frequently observed among cancer patients [13]. Therefore, in this study, using the pre-specified psychoneurological symptom cluster (i.e., pain, fatigue, sleep disturbance, depression), we applied latent class analysis approach to identify distinct patient subgroups characterized by unique symptom cluster profiles. Recognition of symptom profiles in lung cancer survivors can result in better symptom assessment, leading to improved health outcomes that leverage preventative medical practices and more efficient use of limited resources through cluster-targeted treatments [14].

The majority of the relevant studies targeted general cancer survivors, survivors of other types of cancer, or patients with lung cancer [4,15,16]. There have been limited studies focusing on symptom profiles in lung cancer survivors. The purpose of the study was to 1) identify latent class symptom profiles in lung cancer survivors based on four highly prevalent symptoms (i.e., pain, fatigue, sleep disturbance, depression); 2) determine demographic, socioeconomic, and clinical factors that influence the symptom profiles; and 3) examine if these symptom profiles differed on physical function and social function.

This study was guided by the Theory of Unpleasant Symptoms [17]. The theory revolves around the acknowledgement that symptoms are multidimensional, and they tend to be interrelated with each other [17,18]. This recognition allows holistic examination of the symptom experience in relation to influencing factors and performance outcomes. The Theory of Unpleasant Symptoms about symptom profiles and influencing factors in lung cancer survivors is illustrated in Fig 1, showcasing interrelated interactions between socioeconomic/demographic and clinical influencing factors that impact multiple co-occurring symptoms, which in turn impacts performance outcomes.

## Materials and methods

### Samples and setting

We conducted a cross-sectional secondary analysis using data from the Measuring Your Health (MY-Health) study, which aimed to evaluate health and well-being across multiple race-ethnic and age groups of a diverse cohort of cancer patients [19]. Data were obtained from a community-based cancer registry-linked survey. Participants were recruited through four Surveillance, Epidemiology and End Result (SEER) registries located in California, Louisiana, and New Jersey between 2010 and 2012. Participants were eligible for the study if they (a) were aged 21–84 years at the time of initial diagnosis of one of seven types of cancer (i.e., female breast cancer, prostate cancer, colorectal cancer, non-small cell lung cancer, non-Hodgkin lymphoma, uterine cancer, and cervical cancer) and (b) had the ability to read and speak English, Spanish, or Mandarin. The survey was completed by 5,506 people with cancer. For this analysis, the authors restricted eligibility to individuals diagnosed with non-small cell lung cancer within the past 6–13 months. Of 722 patients with lung cancer, a total of 526 participants were included following the exclusion of 35 participants diagnosed more than 13 months prior to the study and 161 participants who had died during the survey period. Details of the study design, study procedures, and participant descriptions of MY-Health study have been reported elsewhere [19].

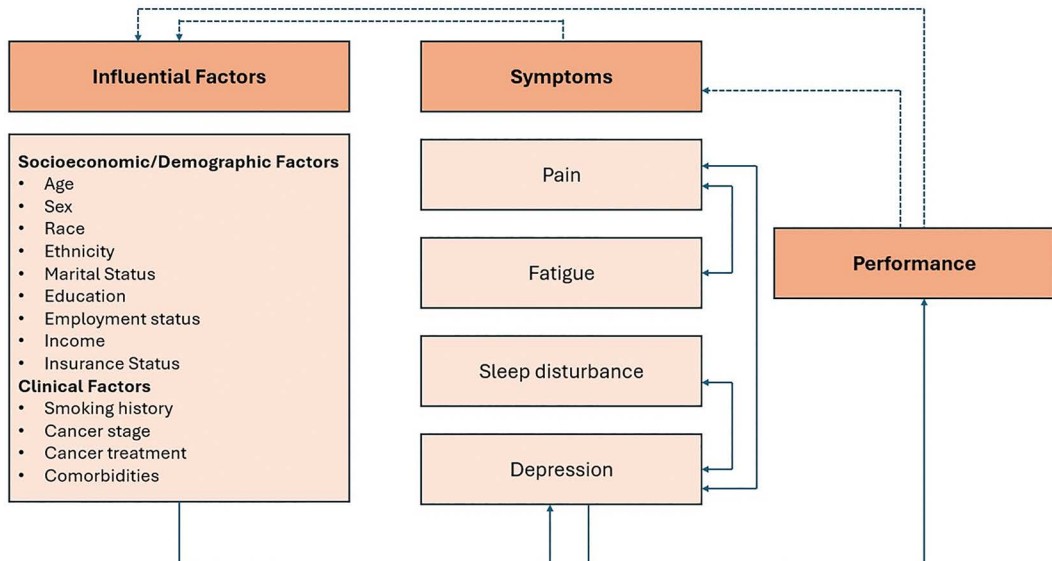

**Fig 1. The Theory of Unpleasant Symptoms Adapted for the Analysis of Symptom Profiles in Lung Cancer Survivors.** Note. Dashed lines correlate to feedback, solid lines correlate to influences, and bi-directional arrows correlate to symptom interaction.

## Measures

**Influential Factors.** Demographic and socioeconomic factors included age at diagnosis, sex, race (white, non-white [multiple, black, Asian, American Indian or Alaska Native, Asian Hawaiian or Pacific Islander, other]), ethnicity (Hispanic, non-Hispanic), marital status (married [married, living with partner], not married [never married, separated, divorced, widowed]), education level (≤ high school [<high school graduate, high school graduate], ≥ some college [some college, college degree, graduate degree]), employment status (working [working full time, full time homemaker, part time, student], not working [retired, disability, unemployed]), annual income (<$60K [less than $10,000, $10,000 to $59,999], ≥ $60K [$60,000 to $99,999, $100,000 to $199,999, $200,000 or more]), and insurance status (private [private, private and government], not private [government only, no insurance]). Clinical factors included smoking history (yes, no), cancer stage (I, II, III, IV), cancer treatment history (no treatment, surgery only, chemotherapy only, radiation therapy only, combination therapy), and the number of self-reported comorbidities (0, ≥ 1).

**Symptoms.** Symptoms were measured using the Patient-Reported Outcomes Measurement Information System (PROMIS®), which have been extensively validated in patients with cancer [20]. MY-Health study administered custom PROMIS short forms assessing pain interference (10 items), fatigue (14 items), sleep disturbance (10 items), and depression (10 items). In the present study, the internal consistency of each measure was acceptable (Cronbach's alpha for pain = 0.984; fatigue = 0.960; sleep disturbance = 0.949; and depression = 0.972). The PROMIS measures are scored on a five-point Likert-type scale, rating from 1 (never) to 5 (always), with higher scores indicating greater symptom severity. The PROMIS are calibrated and standardized to a T-score metric, with a mean of 50 and standard deviation of 10 centered on the general U.S population. The PROMIS enables comparisons across studies using other instruments that measure the same health concept, for example, the Center for Epidemiologic Studies Depression Scale (CES-D) and the Functional Assessment of Chronic Illness Therapy (FACIT)-Fatigue.

Reference PROMIS® T-scores for symptoms like pain, fatigue, sleep disturbance, and depression have been established for the cancer patient subpopulation and are as follows [21,22]:

- Pain: < 50 normal; 50–59 mild; 60–69 moderate; ≥ 70 severe

- Fatigue: < 50 normal; 50–54 mild; 55–74 moderate; ≥ 75 severe

- Depression: < 55 normal, 55–64 mild; 65–74 moderate; ≥ 75 severe

- Sleep disturbance: < 45 normal, 45–54 mild; 55–59 moderate: ≥ 60 severe

**Performance.** Physical function was measured using the PROMIS® Physical Function instruments (16 items) which assesses the function of the patient's upper extremities (dexterity), lower extremities (mobility), and central regions (neck, back), along with monitoring the patient's ability to take part in activities to support day-to-day living [20,23]. For the assessment of social function, the PROMIS® Satisfaction with Social Roles and Activities Instrument (10 items, abbreviated as social function hereafter) was used [20,24]. This instrument assesses whether a patient is satisfied with his or her social roles and participation in discretionary social activities (Cronbach's alpha for physical function = 0.956; and social function = 0.978).

## Statistical analysis

Descriptive statistics were used to describe each variable, including mean, standard deviation, frequency, and percentage. Latent profile analysis (LPA), which is latent class analysis when indicators are continuous variables, was conducted to identify subgroups of lung cancer survivors with similar symptom profiles [11]. We performed LPA via the maximum-likelihood estimation with robust standard errors based on four symptom variables: pain, fatigue, sleep disturbance, and depression. A set of model fit indices were used to determine the number of latent classes that best describe the patterns

observed in our data. The Akaike information criterion (AIC) and the Bayesian information criterion (BIC) are values that indicate how well the model predicts the data; smaller is better. Entropy is a measure of class separation, with values closer to 1 preferred and above 0.8 acceptable. The Vuong–Lo–Mendell–Rubin likelihood ratio test (VLMR) and the parametric bootstrapped likelihood ratio test (BLRT) evaluate whether the less restrictive model is better than the more restricted model. We selected the final number of latent classes by comprehensively reviewing all the model fit indices and with the clinical interpretation of the latent classes.

Following LPA, we conducted multinomial logistic regression analysis to determine influencing factors of the latent class membership. Bivariate analyses were performed on each of the following demographic, socioeconomic, and clinical variables: age at diagnosis, sex, race, ethnicity, marital status, education level, employment status, annual income, insurance status, smoking history, cancer stage, surgery, chemotherapy, radiation therapy, and comorbidities. Variables with $p < .10$ in bivariate analyses were included in multivariate analyses. We selected variables in the final model using backward elimination with a removal criterion of $p \geq .10$. Additionally, we conducted a multivariate analysis of variance (MANOVA) along with post hoc tests to investigate whether physical function and social function differ across the latent classes. We used Mplus Version 8.10 for LPA and IBM SPSS Version 29.0 for all other analyses [25,26].

### Ethical considerations

This study employed a secondary data analysis approach using the MY-Health study. All participants provided informed consent before participation, approved by the institutional review boards of each participating SEER site and Georgetown University, Washington, DC.

## Results

### Sample characteristics

Characteristics of our sample are presented in Table 1. The participants were predominantly older at diagnosis (older than 65) (51.7%), female (53.2%), White (69.2%), non-Hispanic (90.3%), married (57.6%), had completed some college or above (56.2%), were not working (77.0%), made less than $60,000 per year (71.6%), had private insurance (64.1%), had a history of smoking (80.0%), received combination therapy (50.4%), and had one or more comorbidities (80.4%).

### Identification of latent class symptom profiles

The results of statistical fit indices for the candidate models are shown in Table 2. As summarized in Table 3 and illustrated in Fig 2, four distinct classes of lung cancer survivors were identified. Latent classes were labeled based on established symptom cut-off points [21,22]. Class 1 (46.4%), labeled *Within Normal Limits (WNL)*, was characterized by all symptoms within normal limits. Class 2 (24.5%) was characterized by presence of mild pain, fatigue, and sleep disturbance, but with depression within normal limits, and was labeled *Pain with Fatigue and Sleep disturbance (PFSD)*. Class 3 (12.9%) was characterized by presence of moderate fatigue, mild sleep disturbance and depression, but with pain within normal limits, and was labeled *Depression with Fatigue and Sleep disturbance (DFSD)*. Finally, Class 4 (16.2%) was characterized by all symptoms within moderate to severe limits and was labeled as *All High Symptom Burden (AHSB)*.

### Influential factors and latent class symptom profiles

Unadjusted models were significant for age at diagnosis, ethnicity, marital status, education, employment status, annual income, insurance status, and the number of comorbidities. These variables were retained in the multinomial logistic regression model. Table 4 displays the multinomial logistic regression results with variables for each class, using Class 1 as the reference group.

**Table 1.** Sample Characteristics (N = 526).

| Characteristics | Total | Class 1 WNL (n = 244) | Class 2 PFSD (n = 129) | Class 3 DFSD (n = 68) | Class 4 AHSB (n = 85) | p-value |
|---|---|---|---|---|---|---|
| | **M (SD), range** | | | | | |
| | **N(%)** | | | | | |
| **Age at diagnosis (years)** | | | | | | |
| 65 and younger | 254 (48.3) | 95 (38.9) | 66 (51.2) | 32 (47.1) | 61 (71.8) | <.001 |
| Older than 65 | 272 (51.7) | 149 (61.1) | 63 (48.8) | 36 (52.9) | 24 (28.2) | |
| **Sex** | | | | | | |
| Male | 246 (46.8) | 116 (47.5) | 53 (41.1) | 30 (44.1) | 47 (55.3) | .221 |
| Female | 280 (53.2) | 128 (52.5) | 76 (58.9) | 38 (55.9) | 38 (44.7) | |
| **Race** | | | | | | |
| White | 364 (69.2) | 172 (70.5) | 90 (69.8) | 48 (70.6) | 54 (63.5) | .670 |
| Non-White | 162 (30.8) | 72 (29.5) | 39 (30.2) | 20 (29.4) | 31 (36.5) | |
| **Ethnicity** | | | | | | |
| Non-Hispanic | 475 (90.3) | 228 (93.4) | 115 (89.1) | 60 (88.2) | 72 (84.7) | .097 |
| Hispanic | 51 (9.7) | 16 (6.6) | 14 (10.9) | 8 (11.8) | 13 (15.3) | |
| **Marital status** | | | | | | |
| Married | 302 (57.6) | 159 (65.2) | 65 (50.8) | 40 (58.8) | 38 (45.2) | .004 |
| Not married | 222 (42.4) | 85 (34.8) | 63 (49.2) | 28 (41.2) | 46 (54.8) | |
| **Educational level** | | | | | | |
| High school or less | 228 (43.8) | 87 (36.0) | 62 (48.8) | 32 (47.8) | 47 (56.0) | .005 |
| Some college or more | 292 (56.2) | 155 (64.0) | 65 (51.2) | 35 (52.2) | 37 (44.0) | |
| **Employment** | | | | | | |
| Working | 119 (23.0) | 67 (27.8) | 25 (19.5) | 19 (28.8) | 8 (9.8) | .004 |
| Not working | 398 (77.0) | 174 (72.2) | 103 (80.5) | 47 (71.2) | 74 (90.2) | |
| **Annual income** | | | | | | |
| Less than 60,000 | 315 (71.6) | 118 (57.3) | 88 (85.4) | 48 (81.4) | 61 (84.7) | <.001 |
| More than 60,000 | 125 (28.4) | 88 (42.7) | 15 (14.6) | 11 (18.6) | 11 (15.3) | |
| **Insurance status** | | | | | | |
| Private | 330 (64.1) | 168 (70.0) | 80 (63.0) | 46 (67.6) | 36 (45.0) | <.001 |
| Not private | 185 (35.9) | 72 (30.0) | 47 (37.0) | 22 (32.4) | 44 (55.0) | |
| **Smoking history** | | | | | | |
| Yes | 419 (80.0) | 188 (77.0) | 109 (85.2) | 53 (77.9) | 69 (82.1) | .276 |
| No | 105 (20.0) | 56 (23.0) | 19 (14.8) | 15 (22.1) | 15 (17.9) | |
| **Cancer stage** | | | | | | |
| I | 185 (36.3) | 92 (39.0) | 45 (36.0) | 22 (33.8) | 26 (31.0) | .842 |
| II | 73 (14.3) | 32 (13.6) | 19 (15.2) | 9 (13.8) | 13 (15.5) | |
| III | 138 (27.1) | 59 (25.0) | 39 (31.2) | 17 (26.2) | 23 (27.4) | |
| IV | 114 (22.4) | 53 (22.5) | 22 (17.6) | 17 (26.2) | 22 (26.2) | |
| **Cancer treatment history** | | | | | | |
| No treatment | 20 (3.9) | 10 (4.1) | 2 (1.6) | 4 (6.3) | 4 (5.0) | .387 |
| Surgery only | 147 (28.4) | 81 (33.2) | 31 (32.5) | 11 (16.9) | 14 (17.1) | .004 |
| Chemotherapy only | 57 (11.1) | 27 (11.1) | 13 (10.3) | 8 (12.5) | 9 (11.3) | .976 |
| Radiation therapy only | 31 (6.0) | 15 (6.1) | 5 (4.0) | 4 (6.3) | 7 (8.8) | .572 |
| Combination therapy | 259 (50.4) | 111 (45.4) | 65 (51.6) | 37 (57.8) | 46 (57.5) | .142 |

*(Continued)*

 

**Table 1.** (Continued)

| Characteristics | Total | Class 1 WNL (n=244) | Class 2 PFSD (n=129) | Class 3 DFSD (n=68) | Class 4 AHSB (n=85) | p-value |
|---|---|---|---|---|---|---|
| | M (SD), range | | | | | |
| | N(%) | | | | | |
| **Number of comorbidities** | | | | | | |
| | 2.20 (1.84) | 1.54 (1.43) | 2.53 (1.97) | 2.37 (1.73) | 3.48 (1.93) | <.001 |
| Zero | 103 (19.6) | 68 (27.9) | 21 (16.3) | 9 (13.2) | 5 (5.9) | <.001 |
| One or more | 423 (80.4) | 176 (72.1) | 108 (83.7) | 59 (86.8) | 80 (94.1) | |

a. Not married includes never married, living with partner, separated, or divorced.

b. Not working includes retired, disabled, or unemployed; working includes full-time employed, part-time student, and full-time homemaker.

c. Private includes combination and private; not private includes government insurance and uninsured.

d. Unknown stage and occult stage data was not included.

e. Participants could choose multiple responses.

f. Comorbidities include heart failure, heart attack, stroke, asthma, lung disease, diabetes, arthritis, and Alzheimer's disease. Participants could choose multiple responses.

**Table 2. Model Fit Information of LPA Models.**

| Class | Number of parameters | AIC | BIC | Entropy | VLMR[a] | BLRT[a] |
|---|---|---|---|---|---|---|
| 2 | 13 | 14325.912 | 14381.361 | 0.867 | p<.001 | p<.001 |
| 3 | 18 | 14163.871 | 14240.647 | 0.805 | p=0.0189 | p<.001 |
| 4[b] | 23 | 14015.976 | 14114.078 | 0.853 | p=0.0149 | p<.001 |
| 5 | 28 | 13973.228 | 14092.657 | 0.857 | p=0.2992 | p<.001 |

Abbreviations: AIC=Akaike information criterion; BIC=Bayesian information criterion; VLMR=Vuong-Lo-Mendell-Ruben likelihood ratio test; BLRT=bootstrap likelihood ratio test.

[a] Chi-square statistic for the VLMR and the BLRT; when nonsignificant ($p > .05$), the VLMR and BLRT test.

[b] Four-class model was selected based on its having a smaller BIC than the three-class model and nonsignificant VLMR in the five-class model.

**Table 3. Differences in Symptoms Among the Latent Classes (N=526).**

| | Class 1 WNL (n=244) | | Class 2 PFSD (n=129) | | Class 3 DFSD (n=68) | | Class 4 AHSB (n=85) | | |
|---|---|---|---|---|---|---|---|---|---|
| Variable | Mean | SD | Mean | SD | Mean | SD | Mean | SD | p-value |
| Pain | 42.37 | 3.32 | 57.86 | 4.85 | 44.14 | 4.37 | 65.04 | 4.73 | <.001 |
| Fatigue | 42.07 | 5.94 | 54.69 | 5.66 | 55.08 | 8.22 | 62.51 | 5.24 | <.001 |
| Sleep disturbance | 44.02 | 7.53 | 52.08 | 8.06 | 54.25 | 8.80 | 60.47 | 7.86 | <.001 |
| Depression | 43.04 | 3.39 | 48.53 | 5.94 | 57.82 | 7.43 | 65.41 | 7.23 | <.001 |

Abbreviations: SD=standard deviation; WNL = within normal limits; PFSD=pain with fatigue and sleep disturbance; DFSD=depression with fatigue and sleep disturbance; AHSB=all high symptom burden.

**Comparison of Class 2 (*PFSD*) versus Class 1 (*WNL*).** Age at diagnosis, annual income, and the number of comorbidities were significantly associated with the likelihood of reporting a pain, fatigue, and sleep disturbance-related symptom cluster (Class 2). Patients who were 65 years or younger (in comparison to patients who were older than 65) were more likely to be in *PFSD* versus *WNL* (odds ratio [OR] = 2.721, 95% confidence interval [CI] [1.505,4.918]). Patients

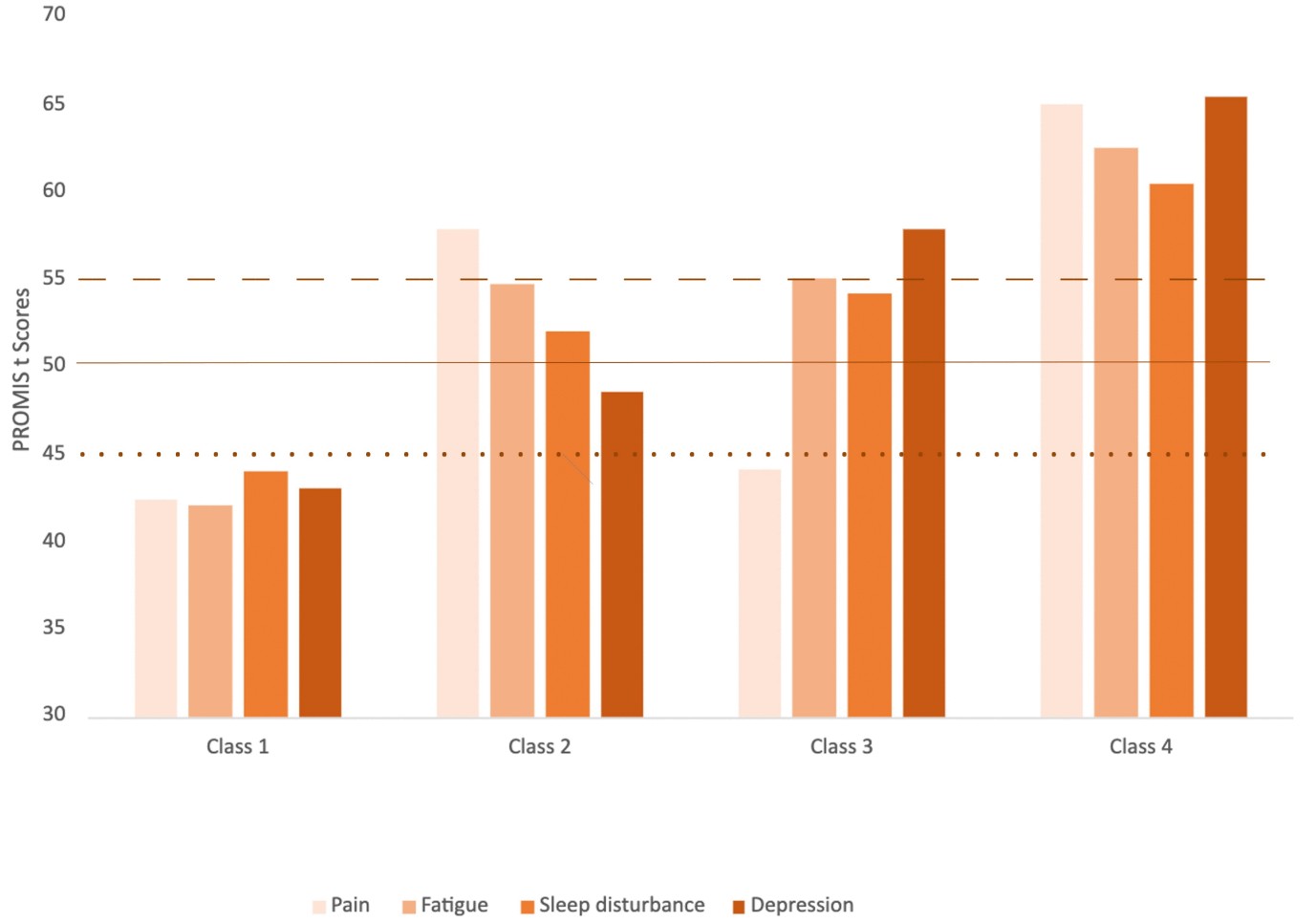

**Fig 2. Difference in Symptoms Among the Latent Classes.** Note. Dashed line at 55 is depression threshold. Solid line at 50 is pain and fatigue threshold. Dotted line at 45 is sleep disturbance threshold.

who had an annual income of less than $60,000 (in comparison to patients who had an annual income of $60,000 or more) were more likely to be in Class 2 versus *WNL* (OR = 3.563, 95% CI [1.787, 7.104]). Additionally, patients with more comorbid conditions were more likely to be in *PFSD* versus *WNL* (OR = 1.353, 95% CI [1.083, 1.692]).

**Comparison of Class 3 (*DFSD*) versus Class 1 (*WNL*).** Annual income was the only significant factor of reporting a depression, fatigue, and sleep disturbance-related symptom cluster (Class 3). Patients who had an annual income of less than $60,000 (in comparison to patients who had an annual income of $60,000 or more) were more likely to be in *DFSD* versus *WNL* (OR = 3.803, 95% CI [1.688, 8.572]).

**Comparison of Class 4 (*AHSB*) versus Class 1 (*WNL*).** Age at diagnosis, employment status, and annual income were significantly associated with the likelihood of reporting a pain, fatigue, depression, and sleep disturbance-related symptom cluster (Class 4). Patients who were 65 years or younger (in comparison to patients who were older than 65) were more likely to be in *AHSB* versus *WNL* (OR = 6.789, 95% CI [3.168, 14.548]). Additionally, patients who were not working (in comparison to those who were working) were more likely to be in *AHSB* versus *WNL* (OR = 3.860, 95% CI [1.383, 10.774]). Patients who had an annual income of less than $60,000 (in comparison to patients who had an annual income of $60,000 or more) were more likely to be in *AHSB* versus *WNL* (OR = 3.371, 95% CI [1.456, 7.804]).

**Table 4. Multinomial Logistic Regression for Latent Class Symptom Profiles.**

| Models | Variables | B | SE | Wald ($\chi^2$) | p-value | OR (95% CI) |
|---|---|---|---|---|---|---|
| Class 2 (PFSD) vs. Class 1 | Intercept | −2.757 | .446 | 38.261 | <.001 | |
| | Age at diagnosis (Ref: >65 years) | 1.001 | .302 | 10.978 | <.001 | 2.721 (1.505, 4.918) |
| | Employment status (Ref: Working) | .076 | 0.341 | 0.050 | .823 | 1.079 (.553, 2.106) |
| | Annual income (Ref: ≥ $60K) | 1.271 | .352 | 13.020 | <.001 | 3.563 (1.787, 7.104) |
| | Number of comorbidities | .303 | .114 | 7.061 | .008 | 1.353 (1.083, 1.692) |
| Class 3 (DFSD) vs. Class 1 | Intercept | −2.697 | .509 | 28.066 | <.001 | |
| | Age at diagnosis (Ref: >65 years) | .571 | .342 | 2.796 | .094 | 1.771 (.906, 3.459) |
| | Employment status (Ref: Working) | .109 | .393 | .077 | .781 | 1.116 (.516, 2.412) |
| | Annual income (Ref: ≥ $60K) | 1.336 | .415 | 10.382 | .001 | 3.803 (1.688, 8.572) |
| | Number of comorbidities | .098 | .133 | .542 | .462 | 1.103 (.849, 1.432) |
| Class 4 (AHSB)vs. Class 1 | Intercept | −4.724 | .672 | 49.361 | <.001 | |
| | Age at diagnosis (Ref: >65 years) | 1.915 | .389 | 24.259 | <.001 | 6.789 (3.168, 14.548) |
| | Employment status (Ref: Working) | 1.351 | .524 | 6.649 | .010 | 3.860 (1.383, 10.774) |
| | Annual income (Ref: ≥ $60K) | 1.215 | .428 | 8.055 | .005 | 3.371 (1.456, 7.804) |
| | Number of comorbidities | .246 | .136 | 3.253 | .071 | 1.279 (.979, 1.670) |

Class 1 (WNL) was used as the reference group.

H0 vs H1: Likelihood ratio $\chi^2$ (df 12) = 73.537 (p <.001).

Nagelkerke Pseudo-$R^2$ = .203 (the model accounts for approximately 20.3% of the total variance).

## Performance and latent class symptom profiles

MANOVA revealed significant differences in physical function and social function between the latent classes: $F$ (6, 954) = 78.235, $p < .001$; Wilks' Lambda = .449. Each univariate ANOVA also revealed that physical function and social function differed significantly between the four latent classes. Post hoc tests were performed using Bonferroni for physical function and Dunnett T3 for social function (due to violation of the equal variance assumption). Post hoc tests showed that: *PFSD*, *DFSD*, and *AHSB* had lower mean scores for physical function and social function than *WNL*; *AHSB* had lower mean scores for physical function and social function than *PFSD* and *DFSD* (see Table 5).

## Discussion

This study identified latent class symptom profiles in lung cancer survivors based on four symptoms (pain, fatigue, sleep disturbance, and depression). It determined the associations of the symptom profiles with influencing factors and physical/social functions. Four distinct classes were identified: Class 1 (*WNL*), Class 2 (*PFSD*), Class 3 (*DFSD*), and Class 4 (*AHSB*). As there has been limited research investigating symptom profiles in lung cancer survivors, this study helps us understand the dynamic nature of multiple concurrent symptoms in lung cancer survivors. This study is novel in that we conducted LPA to identify homogeneous subgroups of lung cancer survivors sharing similar symptom experiences using PROMIS scores, a psychometrically robust measurement system, as indicators.

Previous studies investigating symptom profiles in cancer patients and survivors often include all cancer types [13,15,27–33]. However, relevant studies on lung cancer patients, particularly lung cancer survivors, have been rare [34–37]. These studies varied in the number (ranging from 1 to 6), and type of symptom clusters identified. Limitations across studies may be attributed to differences in the number of symptoms assessed, the assessment tools (e.g., Short Form 36 Health Status Survey, MD Anderson Symptom Inventory), and analytical methods (e.g., factor analysis, cluster analysis, LPA). Despite these variations, two common subgroups have been consistently observed across studies that examined the same pre-specified psychoneurological symptom cluster as in our study (pain, fatigue, sleep disturbance, and depression) in diverse oncology populations, including lung cancer patients, using cluster analysis or LPA: an *all low*

**Table 5. Differences in Physical and Social Function Between Latent Classes.**

| Variables | M (SD) | | | | | p | Post hoc tests |
|---|---|---|---|---|---|---|---|
| | Total sample (N = 482) | Class 1 (WNL) (n = 229) | Class 2 (PFSD) (n = 115) | Class 3 (DFSD) (n = 60) | Class 4 (AHSB) (n = 78) | | |
| Physical function | 50.12 (9.98) | 56.34 (7.66) | 46.14 (7.39) | 48.24 (8.69) | 39.19 (6.92) | <.001 | Class 2, 3, 4 < Class 1; Class 4 < Class 3; Class 4 < Class 2 |
| Social function | 50.19 (9.98) | 57.20 (6.66) | 45.92 (6.84) | 47.97 (8.71) | 37.61 (5.08) | <.001 | Class 2, 3, 4 < Class 1; Class 4 < Class 3; Class 4 < Class 2 |

In post hoc tests, Bonferroni was used for physical function, and Dunnett T3 was used for social function.

subgroup (within normal limits) and an *all high* subgroup [13,28,31–33]. Building on this evidence, our study further highlights the prevalence of these two common subgroups (*all low* and *all high*) among lung cancer survivors.

The results of this study revealed which lung cancer survivors may be at risk for higher symptom burden. Lung cancer survivors who were 65 years or younger were more likely to be part of symptomatic classes (*PFSD* and *AHSB*) in comparison to their older counterparts. Younger age was also significantly associated with higher symptom burden in survivors of other types of cancer, including breast and colorectal cancer [27,38,39]. Although further research is needed on the age-related difference, younger patients are more likely to receive aggressive treatment as they often have a more advanced diagnosis and fewer comorbidities compared to their older counterparts [40]. In addition, older survivors may have relatively lower expectations about their social and vocational roles than younger survivors and may consider their symptoms as part of the aging process [38,41]. Special attention should be given to younger patients' symptom experiences after cancer treatment.

The number of comorbidities was positively associated with the likelihood of being in *PFSD* also increased. It is consistent with the findings of previous studies reporting that having comorbidities could lead to a higher symptom burden in cancer survivors [39,42]. Our study suggests that the presence of comorbid conditions is a key clinical predictor when assessing symptom burden in lung cancer survivors. Comorbidities commonly reported in individuals with lung cancer history include pulmonary diseases (e.g., COPD, asthma), cardiovascular diseases (e.g., heart failure, peripheral vascular disease), diabetes, and renal disease [43]. Health care providers should pay close attention to symptoms experienced by lung cancer survivors who have any comorbid conditions.

Unemployed status and annual income of less than $60,000 were also risk factors for symptom burden in lung cancer survivors. Previous studies have reported significant employment and income losses among lung cancer survivors [44]. The relationship of symptom burden with employment status and income level may be bi-directional. Multiple co-occurring symptoms after cancer treatment could affect survivors' ability to and attitudes toward work and increase the caregiver burden for family members, which could lead to losses in household income [45]. On the one hand, unemployed status and financial problems could worsen these symptoms as survivors may have difficulty maintaining a healthy lifestyle or getting the appropriate follow-up care they need [46,47]. A longitudinal investigation of the association between symptom burden and social determinants of health in lung cancer survivors is recommended.

As for the performance outcomes, levels of physical function and social function were highest in *WNL* and lowest in *AHSB*. This finding is consistent with research indicating that higher symptom burden is associated with functional impairment in older patients with cancer [48] and advanced breast cancer patients, [49] but limited evidence exists in lung cancer survivors. Furthermore, existing literature illustrates how a myriad of factors, such as decreased physical functioning, adverse side-effects resulting from treatments, poor mental health, economic hardship, perceived stigma, and changes in social networks play a role in decreased social functioning for cancer survivors [50]. The authors also found via a

cross-sectional analysis that for cancer survivors, there were larger adverse changes in social well-being if the survivors were women, in younger age categories, belonged to minority groups, and had a lower income [50]. This study adds to the evidence that there could be differences in physical function and social function in lung cancer survivors depending on their symptom experience.

There is limited evidence regarding symptom clusters in lung cancer survivors, while the notion that symptoms in survivors of other types of cancer, such as breast cancer, gastrointestinal cancer, occur in a cluster orientation is well-known and documented. Given the prevalence of lung cancer, especially in the United States, it is important to highlight not only factors influencing symptom cluster membership for lung cancer survivors, but also how these symptoms impact physical and social performance/function for those impacted. It is also important to note that the cross-sectional nature of this study makes it difficult to identify the causal directions for the highlighted symptom clusters, as sleep, fatigue, pain, and depression can be attributed to a plethora of factors such as treatment, underlying disease, and socioeconomic factors.

There are a few relevant limitations of this study. In that vein, we note that the SEER areas cover approximately 30% of the U.S. population and aims to reflect the diverse demographics of the U.S. population. While the registry areas are not considered statistically representative of the U.S. population, studies have shown no significant differences in demographics (age, race, or sex) [51]. However, this U.S. based data analysis cannot be generalized to symptom cluster trends on a global level due to the limitations posed by the U.S. being an outlier when it comes to healthcare costs and associated social issues. In addition, the MY-Health data does not mention any geographical data (rural, urban, etc.), so it limits our understanding of differences and similarities in symptom experience for lung cancer survivors in different parts of the United States. Second, due to the self-reported nature of the data, it is possible that some biases were introduced in terms of participatory willingness to complete the surveys. Similarly, symptoms are self-reported, which can result in some inconsistencies in the data. Finally, due to the fact that this investigation in symptom cluster membership for lung cancer survivors is not longitudinal, the study does not capture necessary changes over time that can provide insights into the dynamic nature of living with cancer.

Regardless of the study's limitations, there are a few strengths that can be highlighted. Firstly, the high statistical significance of the four-class solution builds trust in the efficacy of the study design and findings. Furthermore, this study illustrates the importance of self-reported PROMIS measures and a LPA for processing data to create findings that inform future research regarding symptom cluster presence and membership for lung cancer survivors. The accuracy of the study is supplemented by the high internal consistency of the instrument, along with the study's results following expected predictions based on the findings of other similar studies.

Future research can give us greater insight into different performance measures and how they are impacted by symptom cluster membership. As such, future studies could examine social and physical function for different cancer types as a means to create a comparison model. Additionally, future studies could expand upon other influencing factors (specific insurance types, detailed clinical history, etc.) to understand how other social determinants of health impact symptom cluster membership. In terms of future clinical research, one study focusing on breast cancer patients assessed the relationship between diet and inflammation given its role in psychoneurological symptom clusters; while the results were unclear regarding whether dietary changes can reduce symptoms for survivors [52], it may be valuable to assess these relationships for lung cancer patients.

## Implications for public health

The results of this study have implications for the field of public health due to the significance of socioeconomic and demographic information in governing system cluster membership. This reality parallels a central tenet of public health dialogue: social determinants of health. By understanding the role of such determinants in impacting symptom burden for lung cancer survivors, we can leverage epistemological changes in symptom management protocols to advance patient quality of life. Additionally, such an understanding will be instrumental in preventative medicine strategies to reduce the impact of social determinants on human health systems.

## Conclusion

This study utilized an LPA in order to identify symptoms profiles for lung cancer survivors. The results of this analysis contribute to our understanding of how different socioeconomic, demographic, and clinical factors influence symptom class membership, along with revealing insights into which patient populations are at greatest risk for high symptom burdens. This understanding supplements existing knowledge regarding preventative medicine and the importance of considering social determinants of health and symptom class membership when developing symptom management protocols.

## Author contributions

**Conceptualization:** Tehreem Hussain, Elisa H. Son, Lena J. Lee.

**Formal analysis:** Tehreem Hussain, Elisa H. Son, Li Yang, Lena J. Lee.

**Funding acquisition:** Lena J. Lee.

**Investigation:** Tehreem Hussain, Elisa H. Son, Lena J. Lee.

**Methodology:** Li Yang.

**Supervision:** Gwenyth R. Wallen, Lena J. Lee.

**Validation:** Tehreem Hussain, Elisa H. Son, Gwenyth R. Wallen, Li Yang, Lena J. Lee.

**Visualization:** Tehreem Hussain.

**Writing – original draft:** Tehreem Hussain.

**Writing – review & editing:** Elisa H. Son, Gwenyth R. Wallen, Li Yang, Lena J. Lee.

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
