## [Decision Letter · Decision Letter 0]

19 Jan 2025

Dear Dr. Lee,

Thank you for submitting your manuscript to PLOS ONE. After careful consideration, we feel that it has merit but does not fully meet PLOS ONE’s publication criteria as it currently stands. Therefore, we invite you to submit a revised version of the manuscript that addresses the points raised during the review process.

**ACADEMIC EDITOR:  **

We look forward to receiving your revised manuscript.

Kind regards,

Ming-Ching Lee

Academic Editor

PLOS ONE

Journal Requirements:

3. Your abstract cannot contain citations. Please only include citations in the body text of the manuscript, and ensure that they remain in ascending numerical order on first mention.

4. Please ensure that you refer to Figure 2 in your text as, if accepted, production will need this reference to link the reader to the figure.

Reviewers' comments:

Reviewer's Responses to Questions

**Comments to the Author**

1. Is the manuscript technically sound, and do the data support the conclusions?

Reviewer #1: Partly

Reviewer #2: Yes

2. Has the statistical analysis been performed appropriately and rigorously?

Reviewer #1: Yes

Reviewer #2: Yes

3. Have the authors made all data underlying the findings in their manuscript fully available?

Reviewer #1: No

Reviewer #2: Yes

4. Is the manuscript presented in an intelligible fashion and written in standard English?

Reviewer #1: Yes

Reviewer #2: Yes

Reviewer #1: PONE-D-24-32642

Symptom Profiles in Lung Cancer Survivors: A Latent Class Approach

Thank you for asking me to review this paper. The study described is a Latent Class secondary analysis of pre-existing data that sought to classify symptom clusters of lunc=g cancer patients. Four clusters were identified.

The paper addresses an important and current topic in cancer survivorship and care. However, the cross-sectional nature of the study severely limits the conclusions that can be drawn from the work. A second limitation is that the assessments all used the PROMIS system which does not enable comparison with other work in the symptom cluster area of cancer: it is not clear what these assessments actually measure or what their scores indicate in terms of severity or other dimensions. No explanation is given.

There is no information on how the (presumed) subsample of lung cancer patients were derived from the SEER MY-health study, again making interpretation of the results difficult.

The results need to be more clearly presented. There is no information provided on data coding so we do not know what the significant relationship with income means, for example. The referent is > $60K and the ORs are all positive. I’m assuming that the lower income groups have the higher ORs, but it is not discernable from the way the data are presented.

A further problem is that the data are all from the USA, which is an outlier in terms of the costs of the health care system combined with high levels of social problems, and as such does not represent the majority of nations globally. As the social factors are identified as predictors, the results are therefore of limited utility beyond their national boundaries.

Because the data are cross-sectional, the authors are unable to decompose the causal directions for these symptom clusters. It is already well-known that symptoms cluster in cancers. Sleep disturbance may be due to cancer deregulation of circadian function, to anxiety, to pain or, as is often the case, may pre-date the diagnosis and be a lifelong problem. Ditto with depression and we do not know if the pain described is due to cancer, a late-effect of treatment or arthritis, LBP, or other benign conditions. Similarly, low income households are known to report much higher prevalence of depression independent of physical health.

I’m not sure how much this paper actually contributes to further understanding this complex area. However, if the authors can include some caveats regarding the above severe limitations, the data may be of value to U.S. providers.

Reviewer #2: Thank you for the opportunity to review the manuscript entitled "Symptom Profiles in Lung Cancer Survivors: A Latent Class Approach."

Overall, the manuscript covers a critical topic in the cancer continuum, such as survivorship, including this innovative approach regarding symptom management, specifically for lung cancer. It is well-written and original, the methodology seems robust and appropriate, and the results are presented in a clear fashion.

Nevertheless, I have identified minor issues that should be addressed before acceptance. I hope the suggestions and comments enhance the manuscript's clarity and overall impact.

Please see the specific session comments for further details about necessary revisions.

Specific Comments

Abstract: No comments or suggestions

Introduction: No comments or suggestions

Methodology: The authors state that the “Details of the study design, study procedures, and participant descriptions of MY-Health study have been reported elsewhere.” I would request to include the percentage of the 526 individuals represented in the original study and, furthermore, how many lung cancer survivors are estimated in the specific registries and how representative are the included registries considering all registries in SEER database. (page 5)

Results and Discussion: The authors could explore further the limitations regarding the “sample size” representativeness and any biases related to the willingness to participate in the survey. (page 18)

Figures and Tables: Appropriate

Conclusion: No comments or suggestions

Based on my evaluation, I recommend the manuscript for minor revisions before acceptance.

**Do you want your identity to be public for this peer review?** For information about this choice, including consent withdrawal, please see our Privacy Policy

Reviewer #1: No

Reviewer #2: No

---

## [Author Response · Author response to Decision Letter 1]

3 Mar 2025

Please see the attached responses to reviewers.

---

## [Decision Letter · Decision Letter 1]

4 Jun 2025

Dear Dr. Lee,

Thank you for submitting your manuscript to PLOS ONE. After careful consideration, we feel that it has merit but does not fully meet PLOS ONE’s publication criteria as it currently stands. Therefore, we invite you to submit a revised version of the manuscript that addresses the points raised during the review process.

**Thank you for submitting your manuscript. Please revise your manuscript in accordance with the reviewers' comments and resubmit it for further consideration.**

We look forward to receiving your revised manuscript.

Kind regards,

Made Satya Nugraha Gautama, RN, M.Sc.,M.N.Sc

Academic Editor

PLOS ONE

Reviewers' comments:

Reviewer's Responses to Questions

**Comments to the Author**

Reviewer #3: (No Response)

2. Is the manuscript technically sound, and do the data support the conclusions?

Reviewer #3: Partly

3. Has the statistical analysis been performed appropriately and rigorously?

Reviewer #3: No

4. Have the authors made all data underlying the findings in their manuscript fully available?

Reviewer #3: No

5. Is the manuscript presented in an intelligible fashion and written in standard English?

Reviewer #3: No

**Reviewer #3:**  Please see uploaded document for comments and additional recommendations for revisions. The paper warrants substantial revisions.

**Do you want your identity to be public for this peer review?** For information about this choice, including consent withdrawal, please see our Privacy Policy

Reviewer #3: No

---

## [Author Response · Author response to Decision Letter 2]

27 Sep 2025

Please find attached the responses to the reviewers.

---

## [Editor Report · Decision Letter 2]

17 Oct 2025

Symptom Profiles in Lung Cancer Survivors: A Latent Class Approach

PONE-D-24-32642R2

Dear Dr. Lee,

We’re pleased to inform you that your manuscript has been judged scientifically suitable for publication and will be formally accepted for publication once it meets all outstanding technical requirements.

Kind regards,

Made Satya Nugraha Gautama, RN, M.Sc.,M.N.Sc

Academic Editor

PLOS ONE

---

## [Editor Report · Acceptance letter]

PONE-D-24-32642R2

PLOS ONE

Dear Dr. Lee,

I'm pleased to inform you that your manuscript has been deemed suitable for publication in PLOS ONE. Congratulations! Your manuscript is now being handed over to our production team.

Kind regards,

on behalf of

Mr. Made Satya Nugraha Gautama

Academic Editor

PLOS ONE